# Bioimpedance-assessed volume overload predicts interdialytic hypertension and disrupted circadian blood pressure rhythm in maintenance hemodialysis patients

Qian Wang[1,2], Min Li[1,2], Shipeng Shen[1,2], Yao Hu[1,2], Yongzhe Chen[1,2], Changchang Liang[1,2], Ke Yu[1,2], Ying Li[1,2], Yanqing Chi[1,2☯*], Lu Bai [1,2☯*]

1 Department of Nephrology, Hebei Medical University Third Hospital, Shijiazhuang, Hebei, China,
2 Hebei Key Laboratory of Diabetic Kidney Disease, Shijiazhuang, Hebei, China

☯ These authors contributed equally as corresponding authors.
* 37300757@hebmu.edu.cn (YC); 38700843@hebmu.edu.cn (LB)

## Abstract

### Objective

Hypertension and abnormal circadian blood pressure (BP) rhythm are prevalent in maintenance hemodialysis (MHD) patients, and are closely associated with cardiovascular diseases (CVD) and increased all-cause mortality. Volume overload represents a critical factor in effectively controlling hypertension. Bioelectrical Impedance Analysis (BIA) has been validated as an accurate method for assessing volume status. This study aimed to investigate the predictive value of BIA-derived volume indicators for hypertension and circadian BP rhythm abnormalities in MHD patients, providing a theoretical basis for optimizing volume control and BP management.

### Methods

We used body composition monitor to assess pre-dialysis volume status and employed 44-hour interdialytic ambulatory BP monitoring (ABPM) to obtain BP parameters. Comparative analyses were conducted between controlled vs. uncontrolled ABPM groups, and normal vs. abnormal circadian BP rhythm groups. Univariate and multivariate analyses were performed to identify correlations and independent predictors, with receiver operating characteristic (ROC) curves determining optimal predictive cut-off values.

### Results

Patients in the uncontrolled ABPM group exhibited higher levels of intact parathyroid hormone (iPTH), post-dialysis serum creatinine (Post-HD Scr), post-dialysis urea nitrogen (Post-HD BUN), overhydration (OH), extracellular water (ECW), body

**Data availability statement:** All data generated or analyzed during this study are included in this published article.

**Funding:** This study was supported by the Medical Science Research Project of Hebei under Grant No. 20221164.

**Competing interests:** The authors declare no conflicts of interest related to this study.

surface area-adjusted ECW (ECW/BSA), ECW to total body water ratio (ECW/TBW) and ECW to intracellular water ratio (ECW/ICW), but lower Kt/V and URR. The abnormal circadian BP rhythm group showed higher iPTH, OH, ECW, ECW/BSA and ECW/ICW. Significant correlations were observed: 44h SBP correlated positively with iPTH, OH, ECW/TBW, ECW/ICW, and negatively with Kt/V, URR; 44h DBP correlated positively with iPTH and OH; nocturnal systolic BP decline rate (NSDP) showed negative correlations with iPTH, OH. Multivariate logistic analysis identified OH > 2.35L, ECW/ICW > 0.945 and iPTH > 240.6 pg/ml as independent predictors for hypertension, while OH > 1.55L and iPTH > 203.75 pg/mL predicted abnormal circadian BP rhythm (all $p < 0.05$).

## Conclusion

BIA-derived indicators of volume overload, particularly OH and ECW/ICW, can guide ABPM management in MHD patients and improve long-term outcomes.

## Introduction

Hypertension represents one of the most prevalent complications in maintenance hemodialysis (MHD) patients. Elevated blood pressure (BP) serves as an independent risk factor for the onset and progression of cardiovascular disease (CVD) and increased all-cause mortality [1,2]. Epidemiological data demonstrate an alarmingly high hypertension prevalence of 80–90% among MHD populations, yet BP control rates remain suboptimal at approximately 30% [3–7]. Accurate assessment of BP and effective management of hypertension-related risk factors are crucial for optimizing BP control in MHD patients.

Interdialytic ambulatory BP monitoring (ABPM) has emerged as the gold-standard diagnostic technique, offering simultaneous evaluation of BP control status and nocturnal dipping patterns [8–11]. ABPM better reflects BP levels across various time points and reveals circadian BP rhythm characteristics in MHD patients. It is also more accurate than dialysis-period BP measurements in predicting left ventricular hypertrophy, CVD, and mortality risk [12–15]. In recent years, ABPM has been widely applied in clinical practice and scientific research.

Previous studies have shown that the pathophysiological mechanisms underlying elevated and poorly controlled BP in MHD patients are multifactorial. Factors such as age, coexisting diabetes, volume overload, autonomic dysfunction, negative sodium gradient dialysis, calcium-phosphorus metabolism disorders, secondary hyperparathyroidism (SHPT), and inadequate dialysis all contribute to hypertension and abnormal circadian BP rhythm in these patients to varying degree [16]. Among these, volume overload is considered the most significant and reversible factor [10,17,18]. Consequently, accurate assessment and regulation of volume status remain crucial for BP management in MHD patients. Currently, most medical institutions typically evaluate patients' volume status based on edema severity, urine output, body weight fluctuations, B-type natriuretic peptide (BNP) levels, ultrasound-measured inferior

vena cava (IVC) diameter, and other traditional methods [10,19,20]. However, these methods have limitations such as poor specificity or sensitivity, inadequate reflection of tissue hydration, susceptibility to cardiac function variations, time-consuming procedures, high costs, and limited clinical applicability, thereby failing to accurately reflect patients' optimal hydration status. Bioelectrical Impedance Analysis (BIA) first introduced by Thomasset et al. [21] in the 1960s, provides a non-invasive and user-friendly method for assessing volume status and body composition. This method has been validated by isotope dilution and X-ray absorptiometry and can effectively monitor volume status in both hemodialysis (HD) and peritoneal dialysis (PD) patients [22].

This retrospective observational study employed ABPM to analyze BP levels and circadian BP rhythm patterns in MHD patients at the Hemodialysis Center of Hebei Medical University Third Hospital. Simultaneously, body Composition Monitor (BCM) based on the BIA principle was utilized to assess volume status. The aim was to clarify the characteristics of BP patterns in MHD patients and their influencing factors, particularly BIA-derived volume overload, providing actionable evidence for improving hypertension control and CVD outcomes.

## Subjects and methods

### Study participants

This retrospective observational study was conducted at the Hemodialysis Center of Hebei Medical University Third Hospital from August 1, 2022 to January 15, 2024. Inclusion criteria: ① MHD for at least 3 months (2–3 sessions/week, 4-5h/session), with using low-flux HD, dialysate temperature at 36.5°C, bicarbonate-based dialysate (Na$^+$ 138 mmol/L, K$^+$ 2 mmol/L, Ca$^{2+}$ 1.25 mmol/L), and all using autologous arteriovenous fistula as the HD access; ② Age between 18 and 80 years; ③ Stable dry weight (< 2% variation) for 2 weeks pre-enrollment; ④ No antihypertensive medication adjustments for 2 weeks pre-enrollment. Exclusion criteria: ① Acute renal failure or progressive renal dysfunction; ② Concurrent PD or kidney transplantation; ③Severe cardiovascular, pulmonary, cerebral or hepatic conditions(e.g., heart failure, pleural effusions, abdominal effusions...); ④ Presence of metallic implants (e.g., stents, pacemakers, artificial joints...); ⑤ Pregnant or lactating women; ⑥ Amputees; ⑦ Patients with severe infections, malignancies, pheochromocytoma, renal artery stenosis, or primary aldosteronism; ⑧ Patients with inability for self-care or mental disorders affecting cooperation.

This study was conducted in accordance with the principles of the Declaration of Helsinki. The experimental scheme was approved by the Ethics Committee of Hebei Medical University Third Hospital (W2022-076-1). All study participants provided written informed consent.

### Collection of clinical data

Demographic information was collected for all participants, including gender, age, dialysis duration, primary disease, antihypertensive medication, presence of diabetes mellitus (DM), dry weight, and interdialytic weight gain (IDWG), IDWG was calculated as [(pre-dialysis weight – weight after the previous dialysis)/dry weight] ×100%. Laboratory test results within 1 month before or after ABPM were recorded. Blood samples for all pre-dialysis tests, including hemoglobin (Hb), albumin (ALB), serum calcium (Ca), serum phosphorus (P), serum intact parathyroid hormone (iPTH), pre-dialysis serum creatinine (Pre-HD Scr) and urea nitrogen (Pre-HD BUN), were drawn before the first dialysis session of the week (i.e., at the end of the long interdialytic interval). Post-dialysis serum creatinine (Post-HD Scr) and urea nitrogen (Post-HD BUN) were measured immediately after that same dialysis session. Dialyzer urea clearance×time/distribution volume (Kt/V) and urea reduction ratio (URR) were calculated to assess dialysis adequacy. The formula were as follows: Kt/V = -Ln(Post-HD BUN/Pre-HD BUN-0.008 × t) + (4–3.5 × Post-HD BUN/Pre-HD BUN)×UF/W, where Ln represents natural logarithm; t denotes dialysis duration in hours; UF stands for ultrafiltration volume in liters; W indicates post-dialysis body weight in kilograms; URR= (Pre-HD BUN – Post-HD BUN)/Pre-HD BUN.

## Volume status assessment

BCM (Fresenius Medical Care, Germany) was used to measure the body composition of participants before the first dialysis session of the week, yielding indicators of volume status such as total body water (TBW), extracellular water (ECW), intracellular water (ICW), and overhydration (OH), from which the ratios ECW/TBW and ECW/ICW were derived. Subsequently, body surface area (BSA)-adjusted parameters, namely ECW/BSA, ICW/BSA, and TBW/BSA, were also calculated, with BSA (m²) determined using the formula: √[height (cm) × weight (kg)/ 3600] [23].

## Ambulatory blood pressure monitoring and parameters

A 44-hour interdialytic ABPM was performed using CMS06C monitors (Jinco, Beijing). The monitoring commenced immediately after the same dialysis session during which the BCM assessment was performed, and continued for 44 hours until the start of the next dialysis session. After resting quietly for 30 minutes, participants wore a properly fitted cuff on the non-fistula arm. Measurements were taken every 30 minutes during daytime hours (6:00–22:00) and hourly during nighttime hours (22:00–6:00). During blood pressure measurement, participants were required to cease physical activity, with the elbow positioned at the same level as the heart. All participants were instructed to maintain their normal daily activities along with IDWG. Measurements were involved in the analysis if > 80% of the recordings were valid, and ≤2 non-consecutive day-hours with < 2 valid measurements, and ≤ 1 nighttime hour without valid recordings per 24-hour period.

The following parameters were obtained: 44-hour mean systolic BP (44h SBP), 44-hour mean diastolic BP (44h DBP), daytime mean systolic BP (dSBP), daytime mean diastolic BP (dDBP), nighttime mean systolic BP (nSBP), and nighttime mean diastolic BP (nDBP). Calculate the nocturnal systolic BP decline rate (NSDP), where NSDP = (dSBP - nSBP)/dSBP×100%.

Based on 44-hour mean arterial pressure measurements and following the 24-hour ABPM threshold of 130/80 mmHg, which is consistent with the 2024 ESC and 2019 JSH guideline [24,25], patients were divided into two groups: controlled ABPM group (44h SBP<130 mmHg and 44h DBP<80 mmHg) and uncontrolled ABPM group (44h SBP≥130 mmHg or 44h DBP≥80 mmHg). The circadian BP rhythm pattern was classified according to NSDP as follows: dipper (NSDP≥10% and < 20%), non-dipper (NSDP≥0 and < 10%) and reverse-dipper (NSDP<0). The dipper pattern was classified as the normal circadian BP rhythm group, while the non-dipper and reverse-dipper patterns were classified as the abnormal circadian BP rhythm group.

## Ethical statement

This experimental scheme was approved by the Ethics Committee of Hebei Medical University Third Hospital (No. W2022-076-1). This study was conducted in accordance with the principles of the Declaration of Helsinki. All renal tissue specimens were obtained with the consent of the subjects or their authorized family members.

## Statistical methods

Data processing and statistical analysis were performed using SPSS 26.0 software (IBM Corp, Armonk, NY, USA) and GraphPad Prism version 10 (GraphPad Software, San Diego, CA, USA). All continuous variables were tested for normality distribution. Normally distributed measurement data were expressed as mean±standard deviation, and comparisons between groups were conducted using independent samples t-test. Non-normally distributed measurement data were expressed as median (lower quartile, upper quartile), and comparisons between groups were performed using the Mann-Whitney U test. Categorical data were expressed as rates or percentages, and intergroup comparisons were analyzed using the chi-square ($\chi^2$) test. Pearson or Spearman correlation analysis was used to determine the correlation between the factors. Variables with $p<0.05$ in univariate logistic regression and a variance inflation factor (VIF) < 5 were included in the multivariable logistic regression model to identify independent predictors. Receiver operating characteristic

(ROC) curve analysis was used to calculate the area under the curve (AUC), and the optimal cut-off value was determined by maximizing Youden's index. Statistical significance was set at p < 0.05 for all tests.

## Results

### Baseline characteristics

The study flowchart is shown in Fig 1. A total of 210 patients from the Third Hospital of Hebei Medical University were evaluated for eligibility, among whom 130 met the inclusion/exclusion criteria and provided written informed consent. After excluding 7 patients due to incomplete ABPM or BCM data, a total of 123 patients with complete datasets were ultimately included in the study, including 57 males and 66 females, with a median age of 61 (49, 69) years and a median dialysis duration of 26 (19, 45) months. Primary renal diseases and their proportions were as follows: chronic glomerulonephritis (55 cases, 44.7%), diabetic nephropathy (34 cases, 27.6%), polycystic kidney disease (7 cases, 5.7%), hypertensive nephropathy (6 cases, 4.9%), lupus nephritis (5 cases, 4.1%), ANCA-associated vasculitis-related kidney damage (3 cases, 2.4%), chronic interstitial nephritis (3 cases, 2.4%), chronic pyelonephritis (3 cases, 2.4%), gout-related kidney damage (2 cases, 1.6%), obstructive nephropathy (2 cases, 1.6%), and unknown etiology (3 cases, 2.4%).

The patients were divided into two groups: the controlled ABPM group (38 cases, 30.9%) and the uncontrolled ABPM group (85 cases, 69.1%). Chronic glomerulonephritis was the most common primary disease in both groups. No statistically significant differences were observed between the two groups in terms of gender distribution, age, dialysis duration, dry weight, IDWG, antihypertensive medication or the number of cases with concomitant diabetes (p > 0.05). Compared with the controlled ABPM group, the uncontrolled ABPM group exhibited lower dipper patterns; higher iPTH, post-HD Scr, and post-HD BUN levels; as well as lower Kt/V and URR (p < 0.05). However, no statistically significant differences were found in Hb, ALB, Pre-HD Scr, Pre-HD BUN, Ca, P or residual renal Kt/V between the two groups (p > 0.05) (Table 1).

Among the 123 patients, 20 cases (16.3%) had normal circadian BP rhythm during the interdialytic period, while 103 cases (83.7%) had abnormal circadian BP rhythm (including 63 non-dippers and 40 reverse-dippers). There were no statistically significant differences between the two groups in terms of gender distribution, age, dialysis duration,

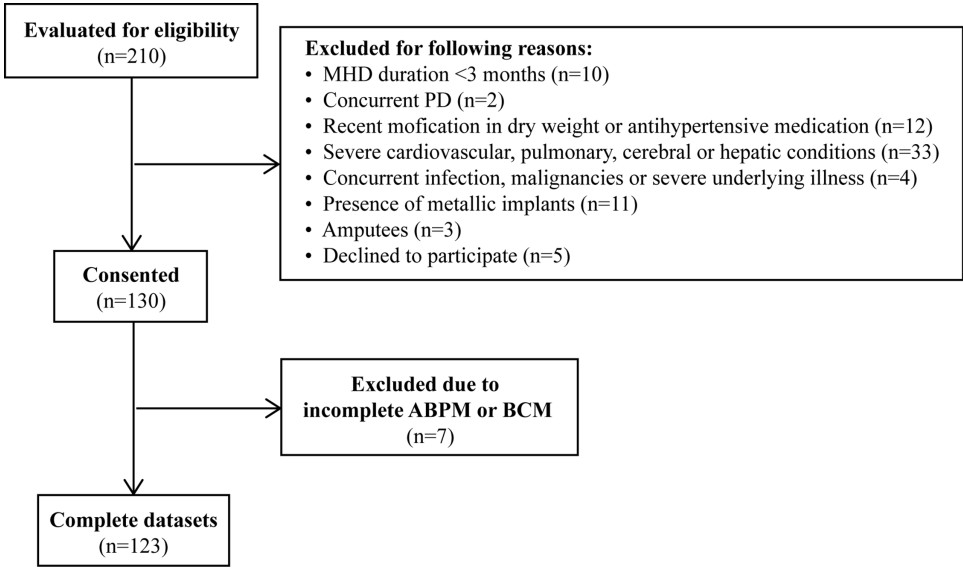

**Fig 1. Flowchart for the study.**

**Table 1. Demographical, clinical, laboratory, and bioimpedance data in controlled vs. uncontrolled ABPM group during the interdialytic period.**

| Variables | Controlled ABPM group (n = 38) | Uncontrolled ABPM group (n = 85) | T/Z/χ² | p |
|---|---|---|---|---|
| Gender(male/female) | 18/20 | 43/42 | 0.109 | 0.741 |
| Age(years) | 59.5 (39, 69) | 61 (50, 69) | −0.282 | 0.778 |
| Dialysis vintage (months) | 24.5 (20, 37) | 26 (19, 56) | −1.233 | 0.217 |
| Weight(Kg) | 59 (48, 62.8) | 60.5 (52, 65.8) | −1.821 | 0.069 |
| IDWG(n, %) | 4.73 (3.72, 6.16) | 4.42 (3.51, 5.44) | −1.199 | 0.230 |
| Combined with diabetes (n, %) | 14 (36.8%) | 30 (35.3%) | 0.027 | 0.869 |
| **Dipper (n, %)** | **10 (26.3%)** | **10 (11.8%)** | **4.083** | **0.043** |
| Total antihypertensive drugs (n) | 2.21 ± 1.38 | 2.40 ± 1.19 | −0.777 | 0.438 |
| ACEis/ARBs (n, %) | 23 (60.5%) | 63 (74.1%) | 2.306 | 0.129 |
| CCBs (n, %) | 21 (55.2%) | 61 (71.1%) | 3.218 | 0.073 |
| β-Blockers (n, %) | 26 (68.4%) | 50 (58.8%) | 1.025 | 0.311 |
| Diuretics (n, %) | 7 (18.4%) | 18 (21.1%) | 0.123 | 0.726 |
| Other antihypertensives (n, %) | 7 (18.4%) | 12 (14.1%) | 0.372 | 0.542 |
| Hb(g/L) | 106 (100, 118) | 106 (97, 117) | −0.463 | 0.643 |
| ALB(g/L) | 40.86 ± 2.93 | 40.85 ± 3.32 | 0.008 | 0.994 |
| Ca(mmol/L) | 2.23 ± 0.15 | 2.22 ± 0.16 | 0.473 | 0.637 |
| P(mmol/L) | 1.87 ± 0.33 | 1.95 ± 0.44 | −0.991 | 0.324 |
| **iPTH(pg/mL)** | **181.2 (149.7, 221.6)** | **210 (156.1, 342.3)** | **−2.663** | **0.008** |
| Pre-HD Scr(μmol/L) | 965.77 (789.58, 1093.33) | 923.11 (803.14, 1154.1) | −0.120 | 0.904 |
| **Post-HD Scr(μmol/L)** | **320.58 (287, 358)** | **334.7 (278.38, 446.54)** | **−2.045** | **0.041** |
| Pre-HD BUN(mmol/L) | 27.12 ± 4.74 | 26.39 ± 5.54 | 0.706 | 0.482 |
| **Post-HD BUN(mmol/L)** | **7.23 (6.1, 8.95)** | **7.97 (6.77, 11)** | **−2.452** | **0.014** |
| **Kt/V** | **1.63 (1.50, 1.77)** | **1.37 (1.18, 1.53)** | **−4.245** | **<0.001** |
| **URR(%)** | **74.03 (70.95, 76.89)** | **67.76 (62.42, 71.89)** | **−3.955** | **<0.001** |
| Residual renal Kt/V | 0.12 ± 0.30 | 0.07 ± 0.16 | 1.068 | 0.291 |

**Abbreviations:** IDWG, interdialytic weight gain; ACEI, angiotensin-converting enzyme inhibitor; ARB, angiotensin receptor blocker; CCBs, calcium channel blockers; Hb, hemoglobin; ALB, albumin; Ca, serum calcium; P, serum phosphorus; iPTH, serum intact parathyroid hormone; Pre-HD Scr, pre-dialysis serum creatinine; Pre-HD BUN, pre-dialysis serum urea nitrogen; Post-HD Scr, post-dialysis serum creatinine; Post-HD BUN, post-dialysis serum urea nitrogen; Kt/V, Calculate dialyzer urea clearance×time/distribution volume; URR, urea reduction ratio. $p < 0.05$ was considered statistically significant.

dry weight, IDWG, antihypertensive medication or the number of patients with diabetes ($p > 0.05$). There were also no differences in 44h SBP and 44h DBP levels between the two groups ($p > 0.05$). Compared with the normal circadian BP rhythm group, the abnormal circadian BP rhythm group had higher iPTH levels and lower Pre-HD BUN levels ($p < 0.05$). Other laboratory parameters including Hb, ALB, Pre-HD Scr, Post-HD Scr, Post-HD BUN, Ca, P, dialysis adequacy (Kt/v, URR) or residual renal Kt/V showed no statistically significant differences between groups ($p > 0.05$) (Table 2).

## Comparison of volume status indicators assessed by BCM

Compared with the controlled ABPM group, patients in the uncontrolled ABPM group exhibited significantly higher values of OH, ECW, ECW/BSA, ECW/TBW and ECW/ICW ($p < 0.05$), while no statistically significant differences were observed in TBW, TBW/BSA, ICW and ICW/BSA between the two groups ($p > 0.05$) (Table 3). Similarly, the abnormal circadian BP

**Table 2. Demographical, clinical, and laboratory data in normal vs. abnormal circadian BP rhythm group during the interdialytic period.**

| Variables | Normal circadian BP rhythm group (n = 20) | Abnormal circadian BP rhythm group (n = 103) | T/Z/χ² | p |
|---|---|---|---|---|
| Gender(male/female) | 9/11 | 52/51 | 0.202 | 0.652 |
| Age(years) | 52(39.5, 65) | 61(51, 69) | −1.775 | 0.076 |
| Dialysis vintage (months) | 31(24, 60.5) | 26(17.5, 44) | −1.301 | 0.193 |
| Weight(Kg) | 54.6(49.9, 61.4) | 60.5(51.2, 65.8) | −1.639 | 0.101 |
| IDWG(%) | 3.87(3.39, 4.76) | 4.66(3.65, 5.57) | −1/522 | 0.128 |
| Combined with diabetes (%) | 5 (25.0%) | 39 (37.9%) | 1.206 | 0.272 |
| 44h SBP (mmHg) | 135.4(124.9, 150.8) | 138.7(129.6, 150.5) | −0.538 | 0.591 |
| 44h DBP(mmHg) | 76.6(73.1, 80.9) | 78.9(71.6, 89.4) | −0.905 | 0.336 |
| Total antihypertensive drugs (n) | 2.40 ± 1.31 | 2.33 ± 1.24 | −0.056 | 0.956 |
| ACEis/ARBs (n, %) | 16 (80%) | 70 (67.9%) | 1.154 | 0.283 |
| CCBs (n, %) | 13 (65%) | 69 (66.9%) | 0.030 | 0.863 |
| β-Blockers (n, %) | 13 (65%) | 63 (61.1%) | 0.104 | 0.747 |
| Diuretics (n, %) | 4 (20%) | 21 (20.3%) | 0.002 | 0.968 |
| Other antihypertensives (n, %) | 2 (10%) | 17 (16.5%) | 0.543 | 0.461 |
| Hb(g/L) | 105(101, 115.5) | 106(97, 118) | −0.298 | 0.765 |
| ALB(g/L) | 39.8(37.8, 41.4) | 41.6(38.7, 43.2) | −1.556 | 0.120 |
| Ca(mmol/L) | 2.21 ± 0.15 | 2.22 ± 0.16 | −0.247 | 0.806 |
| P(mmol/L) | 1.81(1.68, 1.97) | 1.94(1.66, 2.19) | −0.915 | 0.360 |
| **iPTH(pg/mL)** | **153.6(103.1, 194.6)** | **205.2(162.3, 327.9)** | **−3.342** | **0.001** |
| Pre-HD Scr(μmol/L) | 918.49(825.8, 1070.70) | 923.12(794.48, 1154.14) | −0.185 | 0.853 |
| Post-HD Scr(μmol/L) | 327.54(278.38, 379.85) | 332.5(284.24, 427.9) | −0.74 | 0.459 |
| **Pre-HD BUN(mmol/L)** | **29.46(26.78, 32.61)** | **25(22.18, 29.82)** | **−2.183** | **0.029** |
| Post-HD BUN(mmol/L) | 8.74(6.39, 10.82) | 7.4(6.45, 10.56) | −0.661 | 0.508 |
| Kt/V | 1.4(1.3, 1.7) | 1.4(1.2, 1.6) | −0.206 | 0.837 |
| URR(%) | 69.19(66.1, 74.75) | 69.1(64.03, 74.03) | −0.439 | 0.661 |
| Residual renal Kt/V | 0.82 ± 0.18 | 0.85 ± 0.22 | 0.229 | 0.820 |

**Abbreviations:** IDWG, interdialytic weight gain; 44h SBP, 44-hour mean systolic blood pressure; 44h DBP, 44-hour mean diastolic blood pressure; ACEI, angiotensin-converting enzyme inhibitor; ARB, angiotensin receptor blocker; CCBs, calcium channel blockers; Hb, hemoglobin; ALB, albumin; Ca, serum calcium; P, serum phosphorus; iPTH, serum intact parathyroid hormone; Pre-HD Scr, pre-dialysis serum creatinine; Pre-HD BUN, pre-dialysis serum urea nitrogen; Post-HD Scr, post-dialysis serum creatinine; Post-HD BUN, post-dialysis serum urea nitrogen; Kt/V, Calculate dialyzer urea clearance×time/distribution volume; URR, urea reduction ratio. $p < 0.05$ was considered statistically significant.

rhythm group had higher OH, ECW, ECW/BSA and ECW/ICW compared with the normal ABPM rhythm group ($p < 0.05$), whereas other volume status indicators including TBW, TBW/BSA, ICW, ICW/BSA and ECW/TBW showed no significant differences ($p > 0.05$) (Table 4).

## Correlation analysis between ABPM parameters and clinical characteristics or Volume Status Indicators

The study results showed that 44h SBP was positively correlated with iPTH (r = 0.261, $p = 0.004$), Post-HD Scr (r = 0.217, $p = 0.016$), Post-HD BUN (r = 0.266, $p = 0.003$), OH (r = 0.349, $p < 0.001$), ECW/TBW (r = 0.403, $p < 0.001$), ECW/ICW (r = 0.398, $p < 0.001$), and negatively correlated with dialysis adequacy Kt/V (r = −0.299, $p = 0.001$), URR (r = −0.275, $p = 0.002$). 44h DBP was positively correlated with iPTH (r = 0.182, $p = 0.044$), post-HD Scr (r = 0.215, $p = 0.017$), OH (r = 0.233, $p = 0.01$), and negatively correlated with patient age (r = −0.494, $p < 0.001$) (Fig 2). NSDP was negatively correlated with iPTH (r = −0.226, $p = 0.012$) and OH (r = −0.227, $p = 0.011$) (Fig 3).

**Table 3. Bioimpedance data in controlled vs. uncontrolled ABPM group during the interdialytic period.**

| Variables | Controlled ABPM group (n = 38) | Uncontrolled ABPM group (n = 85) | T/Z/χ² | p |
|---|---|---|---|---|
| OH(L) | 1.4 (1.2, 1.9) | 2.4(1.7, 3.3) | −4.882 | <0.001 |
| TBW(L) | 29.4 (28.1, 33.8) | 33.3 (28.3, 37.3) | −1.864 | 0.062 |
| TBW/BSA(L/m²) | 18.19±2.59 | 19.05±3.03 | −1.524 | 0.130 |
| ECW(L) | 14.0 (13.4, 16.3) | 15.6 (14.1, 18.2) | −2.889 | 0.004 |
| ECW/BSA(L/m²) | 8.64±1.24 | 9.37±1.60 | −2.528 | 0.013 |
| ICW(L) | 15.5 (14.6, 17.8) | 16.5 (14.3, 18.6) | −0.706 | 0.480 |
| ICW/BSA(L/m²) | 9.55±1.46 | 9.68±1.67 | −0.408 | 0.684 |
| ECW/TBW | 0.47 (0.46, 0.48) | 0.49 (0.48, 0.51) | −3.731 | <0.001 |
| ECW/ICW | 0.91±0.07 | 0.97±0.12 | −3.935 | <0.001 |

Abbreviations: TBW, total body water; ECW, extracellular water; ICW, intracellular water; OH, overhydration; BSA, body surface area. $p < 0.05$ was considered statistically significant.

**Table 4. Bioimpedance data in normal vs. abnormal circadian BP rhythm group during the interdialytic period.**

| Variables | Normal ABPM rhythm group (n = 20) | Abnormal ABPM rhythm group (n = 103) | T/Z/χ² | p |
|---|---|---|---|---|
| OH(L) | 1.3 (1.1, 1.6) | 2.1 (1.6, 3.2) | −4.076 | <0.001 |
| TBW(L) | 29.05 (26.9, 33.7) | 32.9 (28.5, 36.8) | −1.854 | 0.064 |
| TBW/BSA(L/m²) | 17.70±3.57 | 19.00±2.75 | −1.828 | 0.070 |
| ECW(L) | 13.7 (12.6, 16.1) | 15.6 (14, 17.7) | −2.407 | 0.016 |
| ECW/BSA(L/m²) | 8.47±1.85 | 9.27±1.44 | −2.162 | 0.033 |
| ICW(L) | 15.2 (14.5, 17.6) | 15.8 (14.5, 18.3) | −1.327 | 0.185 |
| ICW/BSA(L/m²) | 9.23±1.44 | 9.72±1.57 | −1.251 | 0.213 |
| ECW/TBW | 0.47 (0.47, 0.49) | 0.49 (0.47, 0.51) | −1.945 | 0.052 |
| ECW/ICW | 0.92±0.07 | 0.96±0.11 | −2.328 | 0.025 |

Abbreviations: TBW, total body water; ECW, extracellular water; ICW, intracellular water; OH, overhydration; BSA, body surface area. $p < 0.05$ was considered statistically significant.

## Analysis of independent predictors for ABPM levels and circadian BP rhythm

To investigate the determinants of interdialytic blood pressure control, univariate and multivariate binary logistic regression analyses were performed. The independent variables included gender, age, dialysis duration, dry weight, IDWG, anti-hypertensive medication, Hb, ALB, Pre-HD Scr, Pre-HD BUN, Post-HD Scr, Post-HD BUN, Kt/V, URR, Ca, P, iPTH, and volume status indicators OH, TBW/BSA, ECW/BSA ICW/BSA, ECW/ICW, ECW/TBW. The dependent variable was ABPM control status. Variables with $p < 0.05$ in univariate analysis and a variance inflation factor (VIF) < 5 were included in the multivariable model. Univariate logistic regression revealed that iPTH, Post-HD Scr, Post-HD BUN, Kt/V, URR, ECW/BSA, ECW/TBW, ECW/ICW, and OH were statistically significant. Among these, URR and ECW/TBW were excluded due to multicollinearity (VIF > 5), the remaining significant variables were entered into the multivariate analysis. The final multivariate analysis identified iPTH, OH and ECW/ICW as independent risk factors for uncontrolled ABPM (Table 5). ROC curve analysis indicated that the AUC values for OH, ECW/ICW and iPTH were 0.776 (95% CI: 0.692–0.860, $p = 0.008$), 0.710 (95% CI: 0.618–0.802, $p < 0.001$) and 0.651 (95% CI: 0.553–0.748, $p < 0.001$), with cut-off values of 2.35 L, 0.945 and 240.6 pg/mL, respectively (Fig 4).

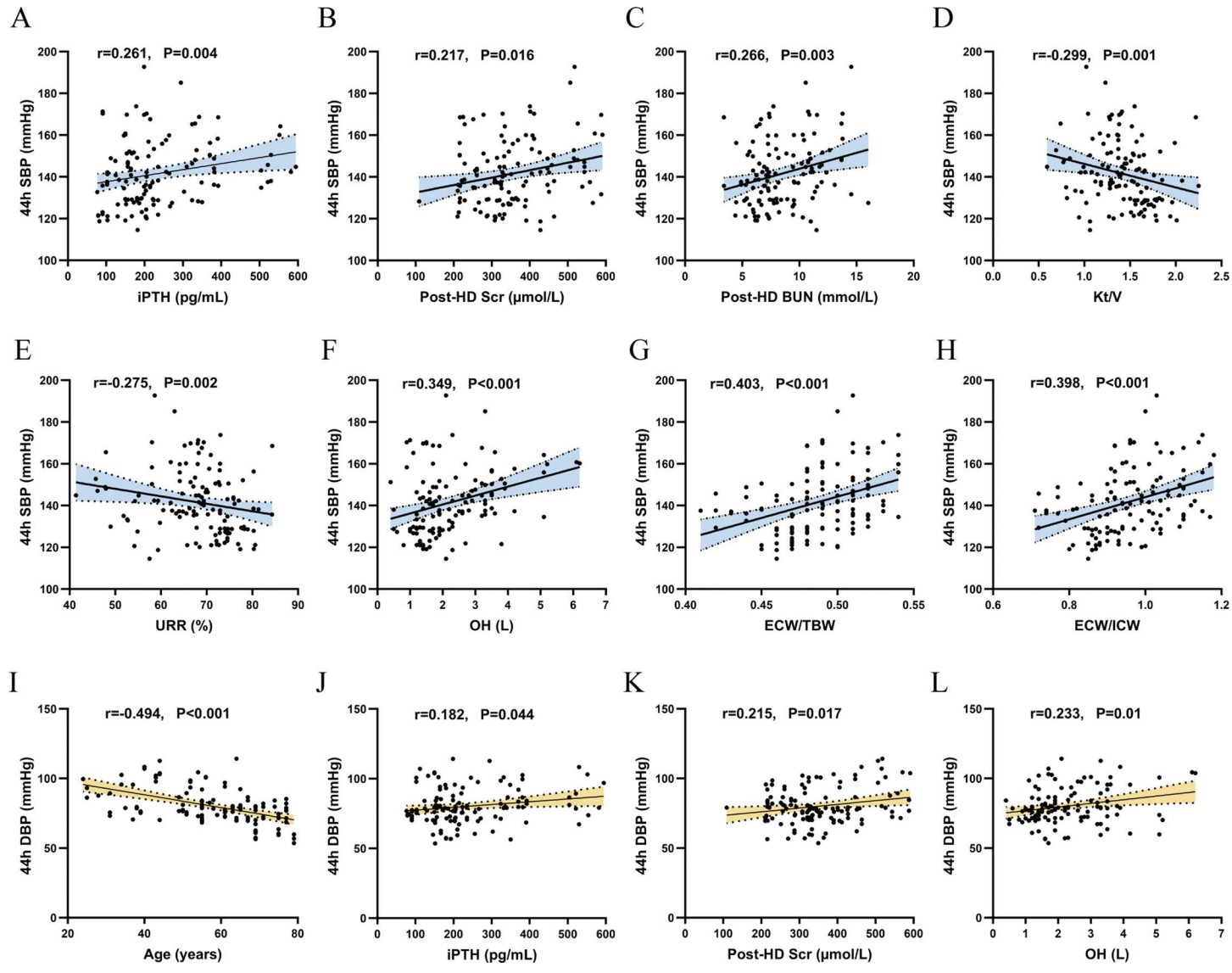

**Fig 2. Correlation analysis of interdialytic blood pressure with clinical, laboratory, and volume status indicators.** Scatter plots showing the correlation between 44h SBP and iPTH **(A)**, Post-HD Scr **(B)**, Post-HD BUN **(C)**, Kt/V **(D)**, URR **(E)**, OH **(F)**, ECW/TBW **(G)**, ECW/ICW **(H)**; 44h DBP with age **(I)**, iPTH **(J)**, Post-HD Scr **(K)**, OH **(L)**. $p<0.05$ was considered statistically significant. **Abbreviations**: 44h SBP, 44-hour mean systolic blood pressure; 44h DBP, 44-hour mean diastolic blood pressure; iPTH, serum intact parathyroid hormone; Post-HD Scr, post-dialysis serum creatinine; Post-HD BUN, post-dialysis serum urea nitrogen; Kt/V, Calculate dialyzer urea clearance×time/distribution volume; URR, urea reduction ratio; TBW, total body water; ECW, extracellular water; ICW, intracellular water; OH, overhydration.

Subsequently, to identify independent predictors of circadian BP rhythm, similar univariate and multivariate logistic regression analyses were conducted, with BP rhythm (normal vs. abnormal) as the dependent variable. Applying the same selection criteria (univariate $p<0.05$, VIF$<5$), variables were screened for inclusion. Univariate analysis identified iPTH, Pre-HD BUN, OH, and ECW/BSA as statistically significant predictors without multicollinearity, and they were thus entered into the multivariate regression. The final model revealed that iPTH and OH were independent risk factors for abnormal circadian BP rhythm (Table 6). ROC curve analysis indicated that the AUC values for OH and iPTH were 0.788

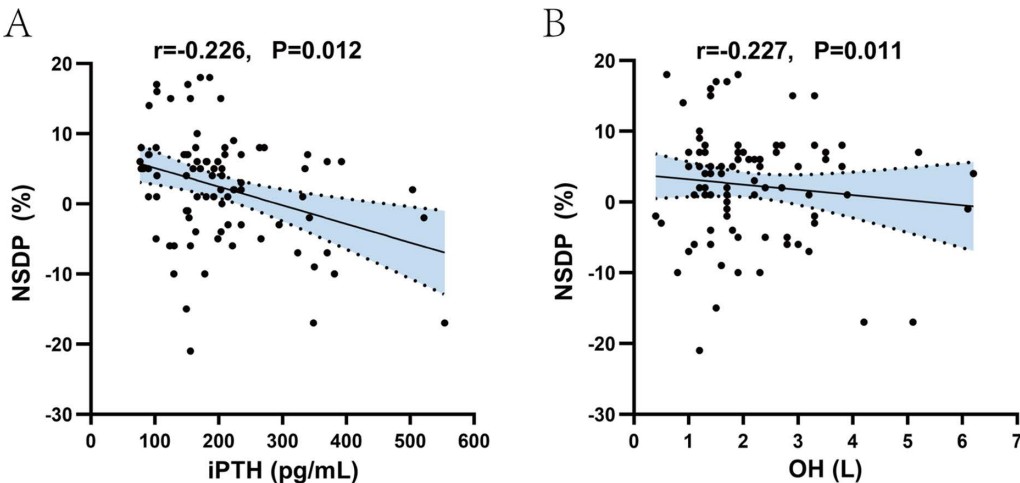

**Fig 3. Correlation analysis of circadian blood pressure rhythm with clinical, laboratory, and volume status indicators.** Scatter plots showing the correlation between NSDP and iPTH (A) and OH **(B)**. *p* < 0.05 was considered statistically significant. **Abbreviations:** NSDP, nocturnal systolic blood pressure decline rate; iPTH, serum intact parathyroid hormone; OH, overhydration.

(95% CI: 0.684–0.892, *p* < 0.001) and 0.737 (95% CI: 0.627–0.846, *p* < 0.001), with cut-off values of 1.55 L and 203.75 pg/mL, respectively (Fig 5).

## Discussion

Hypertension is highly prevalent among end-stage renal disease (ESRD) patients undergoing MHD, and despite the use of multiple antihypertensive medications, BP remains poorly controlled [9,14,26]. In addition to high incidence and treatment challenges, MHD patients also exhibit abnormal circadian BP rhythms characterized by non-dipper or even reverse-dipper patterns. Epidemiological studies show that the incidence of abnormal circadian BP rhythm in MHD patients ranges from 51% to 95% [3,27–29]. In our cohort, the interdialytic hypertension control rate was 30.9%, with an abnormal circadian BP rhythm incidence rate of 83.7%. Given these similar high incidence rate and low control rate, there is an urgent need to identify and manage contributing factors, thereby mitigating CVD risk and all-cause mortality in this population.

Previous studies have shown that various factors can contribute to elevated BP and abnormal circadian BP rhythms in MHD patients. However, volume overload has been consistently identified as the most significant determinant [17]. The increase in volume load in MHD patients can enhance cardiac output and cause inappropriate elevation of vascular resistance through neuroendocrine regulation, thereby triggering or exacerbating hypertension and disrupting circadian BP rhythm [30,31]. In clinical practice, determining the optimal hydration status for MHD patients is highly challenging, and accurately and objectively assessing their volume status and dry weight has been focus research in HD for decades. BIA has emerged as a valuable method for rapid, non-invasive assessment of fluid status. It measures fluid distribution by applying alternating currents at 50 different frequencies (5–1000 kHz), deriving impedance values from high-frequency and low-frequency currents to estimate TBW, ECW and ICW. Using a three-compartment model, it further calculates fat mass (ATM), lean tissue mass (LTM), and OH [32]. Multiple studies have demonstrated that BIA can effectively assess volume status, nutritional status, and prognosis in dialysis patients [33–35]. A clinical study by Moissl et al. [36] involving 55 HD patients confirmed that BIA could effectively guide the optimization of fluid management strategies. Another large-scale study encompassing 39,566 HD patients across 26 countries found that chronic volume overload patients assessed via BIA had a doubled risk of mortality [37]. Pre-dialysis BIA-derived volume overload indicators, such as OH and ECW/

**Table 5. Univariate and multivariable binary logistic regression analysis: clinical and volume-related predictors for uncontrolled ABPM during the interdialytic period.**

| Variables | Univariate analysis | | Multivariate analysis | |
|---|---|---|---|---|
| | OR(95%CI) | p | OR (95%CI) | p |
| Gender(male/female) | 0.977(0.455-2.099) | 0.952 | | |
| Age(years) | 1.005(0.978-1.033) | 0.715 | | |
| Dialysis vintage (months) | 1.012(0.994-1.030) | 0.204 | | |
| Weight(Kg) | 1.040(1.000-1.082) | 0.050 | | |
| IDWG(%) | 0.822(0.668-1.012) | 0.065 | | |
| Total antihypertensive drugs (n) | 1.132(0.829-1.545) | 0.435 | | |
| ACEis/ARBs (n, %) | 1.125 (0.461-2.744) | 0.795 | | |
| CCBs (n, %) | 0.535(0.238-1.206) | 0.131 | | |
| β-Blockers (n, %) | 0.869(0.403-1.873) | 0.720 | | |
| Diuretics (n, %) | 0.841(0.318-2.220) | 0.726 | | |
| Other antihypertensives (n, %) | 0.926(0.365-2.352) | 0.872 | | |
| Hb(g/L) | 0.994(0.971-1.019) | 0.656 | | |
| ALB(g/L) | 1.000(0.886-1.127) | 0.994 | | |
| Ca(mmol/L) | 0.553(0.048-6.337) | 0634 | | |
| P(mmol/L) | 1.635(0.618-4.326) | 0.322 | | |
| iPTH(pg/mL) | **1.007(1.002-1.011)** | **0.003** | **1.006(1.002-1.011)** | **0.007** |
| Pre-HD Scr(μmol/L) | 1.000(0.998-1.002) | 0.899 | | |
| Post-HD Scr(μmol/L) | **1.005(1.001-1.009)** | **0.019** | 1.003(0.995-1.011) | 0.442 |
| Pre-HD BUN(mmol/L) | 0.974(0.906-1.047) | 0.478 | | |
| Post-HD BUN(mmol/L) | **1.210(1.030-1.422)** | **0.021** | 1.013(0.715-1.434) | 0.944 |
| Kt/V | **0.084(0.020-0.359)** | **0.001** | 0.225(0.018-2.735) | 0.242 |
| URR(%) | **0.915(0.864-0.969)** | **0.002** | | |
| Residual renal Kt/V | 0.330(0.060-1.813) | 0.202 | | |
| OH(L) | **3.572(1.939-6.580)** | **<0.001** | **2.941(1.471-5.878)** | **0.002** |
| TBW/BSA(L/m$^2$) | 1.111(0.969-1.273) | 0.131 | | |
| ECW/BSA(L/m$^2$) | **1.408(1.068-1.858)** | **0.015** | 0.930(0.632-1.367) | 0.711 |
| ICW/BSA(L/m$^2$) | 1.052(0.827-1.337) | 0.681 | | |
| ECW/TBW | **1.239(1.071-1.435)** | **0.004** | | |
| ECW/ICW | **1.064(1.023-1.107)** | **0.002** | **1.055(1.001-1.113)** | **0.046** |

**Abbreviations:** IDWG, interdialytic weight gain; ACEI, angiotensin-converting enzyme inhibitor; ARB, angiotensin receptor blocker; CCBs, calcium channel blockers; Hb, hemoglobin; ALB, albumin; Ca, serum calcium; P, serum phosphorus; iPTH, serum intact parathyroid hormone; Pre-HD Scr, pre-dialysis serum creatinine; Pre-HD BUN, pre-dialysis serum urea nitrogen; Post-HD Scr, post-dialysis serum creatinine; Post-HD BUN, post-dialysis serum urea nitrogen; Kt/V, Calculate dialyzer urea clearance×time/distribution volume; URR, urea reduction ratio; TBW, total body water; ECW, extracellular water; ICW, intracellular water; OH, overhydration; BSA, body surface area. $p < 0.05$ was considered statistically significant.

TBW, were positively correlated with patient mortality. Furthermore, OH, OH/ECW, and ECW/TBW were identified as independent risk factors for all-cause mortality in dialysis patients [38,39].

BIA-derived volume management plays a crucial role in managing hypertension and abnormal circadian BP rhythm in MHD patients. A prospective observational study by Furaz Czerpak et al. [40] involving 100 HD patients demonstrated that after BIA-guided dry weight adjustment, significant reductions were observed in OH (P = 0.001), ECW (P = 0.001), TBW (P = 0.036) and ECW/ICW (P = 0.005). Additionally, blood pressure parameters measured via ABPM showed statistically significant decreases, with the greatest reduction in nSBP by 10.36% (±9.94%) and the smallest reduction in mean SBP by 7.04% (±7.97%). Similarly, Nekooeian et al. [41] found that BIA-derived OH was the best predictor of automated office

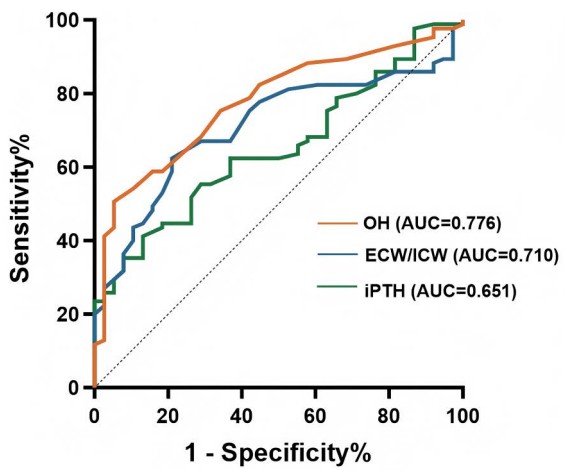

| Variables | Cut-off | AUC | Sen% | Spe% | P | 95% CI | |
|---|---|---|---|---|---|---|---|
| | | | | | | Lower | Upper |
| iPTH(pg/mL) | 240.6 | 0.651 | 41.2 | 86.8 | 0.008 | 0.553 | 0.748 |
| ECW/ICW | 0.945 | 0.710 | 62.4 | 78.9 | <0.001 | 0.618 | 0.802 |
| OH(L) | 2.35 | 0.776 | 50.6 | 94.7 | <0.001 | 0.692 | 0.860 |

**Fig 4. Prediction of uncontrolled interdialytic ambulatory blood pressure.** The receiver operating characteristic (ROC) curves depicting the predictive performance of iPTH, ECW/ICW and OH. $p < 0.05$ was considered statistically significant. **Abbreviations**: iPTH, serum intact parathyroid hormone; ECW, extracellular water; ICW, intracellular water; OH, overhydration.

BP (AOBP) in continuous ambulatory peritoneal dialysis (CAPD) patients. In our study, we utilized BCM, a BIA-based method, to assess volume status in 123 MHD patients, combined with 44-hour interdialytic ABPM. Our results revealed that volume status-related indicators OH, ECW, ECW/BSA and ECW/ICW were significantly elevated in uncontrolled ABPM group and abnormal circadian BP rhythm group. Multivariable analysis identified OH as an independent risk factor for both uncontrolled hypertension and abnormal circadian BP rhythm. Specifically, maintaining pre-dialysis overhydration below 2.35 L may improve blood pressure control, whereas a further reduction below 1.55 L could promote normalization of the circadian BP rhythm. Significant correlations were observed between interdialytic BP parameters and volume indicators: 44h SBP was positively correlated with OH, ECW/TBW, ECW/ICW; 44h DBP was positively correlated with OH; and NSDP was negatively correlated with OH. Interestingly, ECW/ICW and ECW/ICW exhibited stronger correlations with interdialytic BP than OH. This may be attributed to the different physiological information conveyed by these indices. While OH represents absolute fluid overload, ECW/TBW and ECW/ICW reflect the relative distribution of body water. For MHD patients with decreased ICW and reduced physiological buffering capacity due to conditions such as aging, malnutrition, or sarcopenia, even an identical absolute OH value may exert a more pronounced hemodynamic impact, potentially leading OH to underestimate the true degree of fluid overload. Therefore, in this context, ECW/TBW and ECW/ICW may serve as more sensitive indicators for detecting clinically relevant fluid disturbances.

In this study, we also observed variability phenomenon in the relationship between volume status and BP: some patients were hypertension despite low OH, whereas others maintained normal or low BP despite elevated OH. This observation reflected the multifactorial nature of hypertension in MHD patients. Beyond the primary factor of volume overload, other elements such as vascular calcification, inadequate dialysis and SHPT are involved. Accordingly, we systematically analyzed additional demographic, clinical, and laboratory factors that may contribute to uncontrolled hypertension and abnormal circadian BP rhythm in MHD patients. Univariate analysis revealed that the 44h SBP was positively

**Table 6. Univariate and multivariable binary logistic regression analysis: clinical and volume-related predictors for abnormal circadian BP rhythm during the interdialytic period.**

| Variables | Univariate analysis | | Multivariate analysis | |
|---|---|---|---|---|
| | OR(95%CI) | p | OR (95%CI) | p |
| Gender(male/female) | 0.772(0.295-2.020) | 0.598 | | |
| Age(years) | 1.030(0.996-1.064) | 0.087 | | |
| Dialysis vintage (months) | 0.987(0.969-1.005) | 0.153 | | |
| Weight(Kg) | 1.045(0.993-1.099) | 0.089 | | |
| IDWG(%) | 1.150(0.895-1.477) | 0.275 | | |
| Total antihypertensive drugs (n) | 0.956(0.650-1.405) | 0.818 | | |
| ACEis/ARBs (n, %) | 0.564 (0.202-1.574) | 0.274 | | |
| CCBs (n, %) | 1.886(0.585-6.083) | 0.288 | | |
| β-Blockers (n, %) | 2.394(0.8823-6.493) | 0.086 | | |
| Diuretics (n, %) | 0.976(0.295-3.228) | 0.968 | | |
| Other antihypertensives (n, %) | 0.347(0.075-1.599) | 0.174 | | |
| Hb(g/L) | 0.996(0.966-1.026) | 0.777 | | |
| ALB(g/L) | 1.119(0.960-1.305) | 0.150 | | |
| Ca(mmol/L) | 1.475(0.069-31.684) | 0.804 | | |
| P(mmol/L) | 1.471(0.435-4.971) | 0.535 | | |
| iPTH(pg/mL) | **1.011(1.003-1.018)** | **0.005** | **1.009(1.002-1.017)** | **0.019** |
| Pre-HD Scr(μmol/L) | 1.001(0.998-1.003) | 0.665 | | |
| Post-HD Scr(μmol/L) | 1.002(0.997-1.007) | 0.464 | | |
| Pre-HD BUN(mmol/L) | **0.909(0.827-0.999)** | **0.048** | 0.943(0.841-1.057) | 0.312 |
| Post-HD BUN(mmol/L) | 0.937(0.786-1.116) | 0.464 | | |
| Kt/V | 0.842(0.189-3.756) | 0.821 | | |
| URR(%) | 0.984(0.930-1.042) | 0.586 | | |
| Residual renal Kt/V | 1.067(0.110-10.304) | 0.955 | | |
| OH(L) | **4.047(1.741-9.404)** | **0.001** | **3.309(1.294-8.466)** | **0.013** |
| TBW/BSA(L/m²) | 1.175(0.985-1.401) | 0.073 | | |
| ECW/BSA(L/m²) | **1.454(1.024-2.065)** | **0.036** | 1.175(0.804-1.716) | 0.405 |
| ICW/BSA(L/m²) | 1.226(0.889-1.691) | 0.214 | | |
| ECW/TBW | 1.127(0.953-1.333) | 0.163 | | |
| ECW/ICW | 1.039(0.992-1.087) | 0.103 | | |

**Abbreviations**: IDWG, interdialytic weight gain; ACEI, angiotensin-converting enzyme inhibitor; ARB, angiotensin receptor blocker; CCBs, calcium channel blockers; Hb, hemoglobin; ALB, albumin; Ca, serum calcium; P, serum phosphorus; iPTH, serum intact parathyroid hormone; Pre-HD Scr, pre-dialysis serum creatinine; Pre-HD BUN, pre-dialysis serum urea nitrogen; Post-HD Scr, post-dialysis serum creatinine; Post-HD BUN, post-dialysis serum urea nitrogen; Kt/V, Calculate dialyzer urea clearance×time/distribution volume; URR, urea reduction ratio; TBW, total body water; ECW, extracellular water; ICW, intracellular water; OH, overhydration; BSA, body surface area. $p < 0.05$ was considered statistically significant.

correlated with iPTH and negatively correlated with dialysis adequacy (Kt/V and URR); 44h DBP was positively correlated with iPTH. Further multivariate logistic regression analysis indicated that elevated iPTH was an independent risk factor for uncontrolled hypertension and abnormal circadian BP rhythm in MHD patients. Maintaining iPTH < 240.6 pg/mL may assist in achieving better interdialytic BP control, and a further reduction below 203.75 pg/mL could promote normalization of the circadian BP rhythm. These findings align with existing literature reporting that BP variability (BPV) was negatively correlated with Kt/V and CCr but positively associated with hyperphosphatemia and SHPT, and multivariate analysis confirmed that both Kt/V and PTH independently influenced BPV [1]. The observed associations may be explained

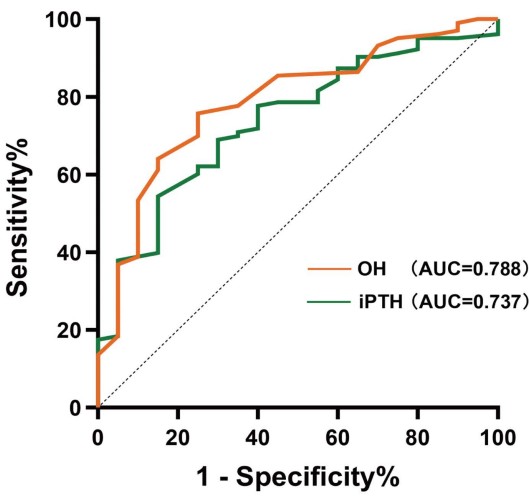

| Variables | Cut-off | AUC | Sen% | Spe% | P | 95% CI | |
|---|---|---|---|---|---|---|---|
| | | | | | | Lower | Upper |
| iPTH(pg/mL) | 203.75 | 0.737 | 54.4 | 85 | <0.001 | 0.627 | 0.846 |
| OH(L) | 1.55 | 0.788 | 75.7 | 75 | <0.001 | 0.684 | 0.892 |

**Fig 5. Prediction of abnormal circadian blood pressure rhythm.** The receiver operating characteristic (ROC) curves depicting the predictive performance of iPTH and OH. *p* < 0.05 was considered statistically significant. **Abbreviations**: iPTH, serum intact parathyroid hormone; OH, overhydration.

through several pathophysiological mechanisms in MHD patients. Elevated PTH levels promote myocardial and vascular calcification, increasing vascular stiffness and resistance, ultimately resulting in hypertension. Additionally, PTH acts as a cardiotoxin by inducing myocardial hypertrophy and fibrosis, which further contribute to BP elevation [42–44]. Concurrently, inadequate dialysis (low Kt/V or URR) contributes to toxin accumulation, calcium-phosphorus metabolic disorders and SHPT, which in turn impair vascular endothelial function and exacerbate hypertension; alternatively, it may induce BP fluctuations and circadian rhythm abnormalities through volume overload and sympathetic activation [33]. Both suboptimal dialysis adequacy and elevated PTH levels independently influence ABPM through distinct pathways, suggesting that clinical management should comprehensively optimize dialysis protocols along with calcium-phosphorus metabolism and SHPT control to improve patient outcomes.

Several limitations of our study should be mentioned. First, our study was conducted at a single center with a relatively modest sample size, which may limit the generalizability of our findings. These results therefore require validation in larger, multicenter studies. Second, the retrospective observational design means that the identified associations do not establish causality. Consequently, our findings need to be confirmed by future prospective, multicenter research that collects more detailed clinical data, which will be essential for exploring the potential causal mechanisms suggested by our results.

## Conclusion

Hypertension and abnormal circadian BP rhythm are common clinical issues in MHD patients, with volume status representing the most critical determinant. This study identified serum iPTH and BIA-derived OH as independent predictors for both hypertension and abnormal circadian BP rhythm. Significant correlations were found between interdialytic BP parameters and iPTH or volume indices, particularly OH and ECW/ICW. Despite the limitations of a single center sample size,

our findings provide clinically reliable evidence for optimizing BP management in MHD patients. Implementing routine BIA assessments to guide precise volume management based on its objective metrics (e.g., OH, ECW/ICW) represents a key actionable strategy to improve hypertension control and may ultimately reduce the risk of CVD and mortality in this population.

## Acknowledgments

We would like to acknowledge all of the physicians and participants from Third Hospital of Hebei Medical University for their support in this study.

## Author contributions

**Conceptualization:** Qian Wang, Lu Bai.

**Data curation:** Qian Wang, Shipeng Shen, Yao Hu, Yongzhe Chen, Changchang Liang, Ke Yu.

**Formal analysis:** Qian Wang.

**Funding acquisition:** Qian Wang, Min Li.

**Investigation:** Qian Wang, Yanqing Chi.

**Methodology:** Qian Wang, Min Li, Yanqing Chi.

**Project administration:** Lu Bai.

**Resources:** Ying Li.

**Software:** Shipeng Shen, Yao Hu, Yongzhe Chen, Changchang Liang, Ke Yu.

**Supervision:** Ying Li, Yanqing Chi.

**Validation:** Lu Bai.

**Visualization:** Yanqing Chi.

**Writing – original draft:** Qian Wang.

**Writing – review & editing:** Min Li, Shipeng Shen, Yao Hu, Yongzhe Chen, Changchang Liang, Ke Yu, Ying Li, Yanqing Chi, Lu Bai.

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
