## [Decision Letter · Decision Letter 0]

10 Dec 2025

PONE-D-25-57307Bioimpedance-assessed volume overload predicts interdialytic hypertension and disrupted circadian blood pressure rhythm in maintenance hemodialysis patientsPLOS One

Dear Dr. Bai,

Thank you for submitting your manuscript to PLOS ONE. After careful consideration, we feel that it has merit but does not fully meet PLOS ONE’s publication criteria as it currently stands. Therefore, we invite you to submit a revised version of the manuscript that addresses the points raised during the review process. 

We look forward to receiving your revised manuscript.

Kind regards,

Yosuke Yamada

Academic Editor

PLOS One

Journal Requirements:

This study was supported by the Medical Science Research Project of Hebei under Grant No. 20221164.

Reviewers' comments:

Reviewer's Responses to Questions

**Comments to the Author**

1. Is the manuscript technically sound, and do the data support the conclusions?

Reviewer #1: Yes

Reviewer #2: Partly

2. Has the statistical analysis been performed appropriately and rigorously? 

Reviewer #1: Yes

Reviewer #2: No

3. Have the authors made all data underlying the findings in their manuscript fully available?

Reviewer #1: Yes

Reviewer #2: Yes

4. Is the manuscript presented in an intelligible fashion and written in standard English?

Reviewer #1: Yes

Reviewer #2: Yes

5. Review Comments to the Author

Reviewer #1: This study is of high clinical relevance as it evaluates the association between fluid overload, abnormal blood pressure, and circadian rhythm disturbances in maintenance hemodialysis patients using a combination of bioelectrical impedance analysis (BIA) and 44-hour ambulatory blood pressure monitoring (ABPM). The results demonstrate that overhydration (OH), intact parathyroid hormone (iPTH), and Kt/V are independent predictors of both hypertension and abnormal BP rhythm, which appears clinically plausible.

I would like to raise the following comments.

1. Definition of hypertension

In this study, hypertension is defined as a 44-hour mean BP ≥130/80 mmHg.

However, the diagnostic thresholds for 24-hour ABPM are not internationally standardized.

For example, ACC/AHA 2017 recommends 125/75 mmHg, whereas ESC/ESH 2023 and JSH 2019 adopt 130/80 mmHg.

Therefore, the authors should clarify the rationale for selecting the 130/80 mmHg cut-off and provide the corresponding guideline or literature reference supporting this choice.

2. Body-size dependency and the need for BSA-adjusted fluid indices

The observed correlation between ECW and blood pressure is notable; however, ECW, ICW, and TBW are strongly influenced by sex and body size.

I recommend presenting BSA-adjusted ECW, ICW, and TBW to improve interpretability.

This is particularly important in maintenance hemodialysis patients, who commonly exhibit reduced ICW due to aging and sarcopenia, which may limit the accuracy of unadjusted ECW values in assessing true volume status.

3. Variability in the relationship between OH and blood pressure

In Figure 2F, several patients show high blood pressure despite low OH, whereas others exhibit low blood pressure despite high OH.

This variability likely reflects the multifactorial nature of hypertension in dialysis patients, involving not only fluid overload but also vascular calcification, autonomic dysfunction, inadequate dialysis, and secondary hyperparathyroidism.

I suggest adding discussion acknowledging this multifactorial background and emphasizing the clinical importance of evaluating fluid overload while also recognizing other contributors to hypertension.

4. Stronger correlations of ECW/TBW and ECW/ICW compared with OH

It is interesting that ECW/TBW and ECW/ICW show stronger correlations with blood pressure compared to OH.

These indices reflect the relative increase in ECW in the context of reduced ICW.

In elderly, malnourished, or sarcopenic patients, the physiological buffer capacity against fluid overload is diminished, and even a similar OH value may exert a greater hemodynamic impact.

Because OH represents an absolute excess fluid volume estimated by BCM, it may underestimate fluid overload in such populations.

From this perspective, ECW/TBW and ECW/ICW may act as more sensitive indices in detecting clinically relevant fluid disturbances.

I encourage the authors to expand the discussion accordingly.

Reviewer #2: General Comment

This study investigates whether bioimpedance-derived volume overload predicts interdialytic hypertension and abnormal circadian blood pressure rhythm assessed by 44-hour ABPM in maintenance hemodialysis patients. The major strength of this work lies in the combined use of BCM-derived volume assessment and prolonged interdialytic ABPM for detailed blood pressure phenotyping.

However, several concerns should be addressed.

Major Comments.

1. Timing of blood sampling, BCM, and ABPM measurements

The timing of blood sampling, BCM measurement, and ABPM initiation is not clearly specified. Please clarify whether blood tests and BCM assessments were uniformly performed at the beginning of the dialysis week and specifically before the first dialysis session of the week. Since volume status and biochemical parameters fluctuate across the dialysis cycle, precise timing is essential for proper interpretation of the relationships between OH, laboratory data, and ABPM results.

2. Statistical methodology.

First, the method used to determine the ROC-derived cut-off values should be clearly specified. Were the cut-off values determined using the Youden Index?

Second, multivariable analyses are presented in Tables 5 and 6. However, the covariates listed in these tables do not fully match all variables described in the main text as being included in the multivariable models. Should the reader understand that all variables listed in the Analysis of factors influencing ABPM levels and circadian rhythm in the Results section were entered into the multivariable models, even if they are not displayed in Tables 5 and 6? This point requires clarification for transparency and reproducibility.

Furthermore, several closely related variables appear to have been entered simultaneously into the models, such as Pre-HD Scr and Post-HD Scr, as well as ECW, ECW/ICW, and ECW/TBW. These variables are mathematically and physiologically interdependent. How was multicollinearity assessed, and how was it addressed in the regression models? Without appropriate handling of multicollinearity, the stability and validity of the reported independent predictors may be compromised.

3. Take-home message and clinical interpretation.

The authors’ main conclusion is that OH is associated with poor blood pressure control and abnormal ABPM circadian rhythm. As this is an observational study, causality cannot be established. However, the Discussion and Conclusion emphasize the concept of “personalized treatment strategies.” Could the authors clarify whether this concept essentially means that reducing pre-dialysis OH—that is, minimizing interdialytic weight gain—is the key practical implication of their findings? Explicitly stating this clinical message would greatly strengthen the practical relevance and interpretability of the study.

Minor Comments.

1. Please spell out the abbreviation NSDP in the Abstract.

2. The manuscript uses multiple terms such as “abnormal circadian rhythm,” “ABPM rhythm,” and “abnormal blood pressure rhythm.” Are these terms intended to represent exactly the same concept? If so, please unify the terminology for consistency. If not, please clarify their respective definitions and distinctions.

6. PLOS authors have the option to publish the peer review history of their article (what does this mean?). If published, this will include your full peer review and any attached files.

Reviewer #1: No

Reviewer #2: **Yes: **Takuya Fujimaru

---

## [Author Response · Author response to Decision Letter 1]

23 Dec 2025

Response to Editors

Response:

Thank you. We have revised the manuscript to comply with PLOS ONE's style and file naming requirements.

2.PLOS requires an ORCID iD for the corresponding author in Editorial Manager on papers submitted after December 6th, 2016. Please ensure that you have an ORCID iD and that it is validated in Editorial Manager. To do this, go to ‘Update my Information’ (in the upper left-hand corner of the main menu), and click on the Fetch/Validate link next to the ORCID field. This will take you to the ORCID site and allow you to create a new iD or authenticate a pre-existing iD in Editorial Manager.

Response:

Thank you for the note. We confirm that the corresponding author has a validated ORCID iD linked to the submission in Editorial Manager.

3.We note that the grant information you provided in the ‘Funding Information’ and ‘Financial Disclosure’ sections do not match. When you resubmit, please ensure that you provide the correct grant numbers for the awards you received for your study in the ‘Funding Information’ section.

Response:

Thank you. The funding information has been verified, and the consistent grant details are now provided in the 'Funding Information' sections (Page 27, Line 472-475).

4. Thank you for stating the following financial disclosure: This study was supported by the Medical Science Research Project of Hebei under Grant No. 20221164. Please state what role the funders took in the study. If the funders had no role, please state: "The funders had no role in study design, data collection and analysis, decision to publish, or preparation of the manuscript." If this statement is not correct you must amend it as needed.Please include this amended Role of Funder statement in your cover letter; we will change the online submission form on your behalf.

Response:

Thank you for your guidance. We confirm that the funders had no role in study design, data collection and analysis, decision to publish, or preparation of the manuscript. This required statement has been included in our revised manuscript (Page 27, Line 472-475).

Response to Reviewer #1

1. Definition of hypertension

In this study, hypertension is defined as a 44-hour mean BP ≥130/80 mmHg. However, the diagnostic thresholds for 24-hour ABPM are not internationally standardized. For example, ACC/AHA 2017 recommends 125/75 mmHg, whereas ESC/ESH 2023 and JSH2019 adopt 130/80 mmHg. Therefore, the authors should clarify the rationale for selecting the 130/80 mmHg cut-off and provide the corresponding guideline or literature reference supporting this choice.

Response:

Thank you for this valuable comment. We have revised the manuscript to explicitly state in the Methods section that the use of the 130/80 mmHg threshold for the 44-hour mean blood pressure aligns with contemporary international guidelines, specifically the 2024 ESC and 2019 JSH guidelines. This clarification ensures clinical relevance and consistency with current hypertension management paradigms. The corresponding references have been added accordingly (Page 8, Line 165-169). 

2.Body-size dependency and the need for BSA-adjusted fluid indices. The observed correlation between ECW and blood pressure is notable; however, ECW, ICW, and TBW are strongly influenced by sex and body size. I recommend presenting BSA-adjusted ECW, ICW, and TBW to improve interpretability. This is particularly important in maintenance hemodialysis patients, who commonly exhibit reduced ICW due to aging and sarcopenia, which may limit the accuracy of unadjusted ECW values in assessing true volume status.

Response:

We sincerely appreciate this constructive suggestion. We concur that the significant body size dependency of TBW, ECW, and ICW is a critical methodological consideration, and that employing BSA-adjusted indices would substantially enhance the clinical interpretability of fluid status assessments. Accordingly, we have calculated BSA-adjusted indices for the core fluid volumes (TBW/BSA, ECW/BSA, ICW/BSA) and integrated them into our analytical framework. The revised manuscript now reports these adjusted indices in conjunction with the original absolute values (Page 7, Line 143-146; Page 14, Line 258-265; Page 16-18, Line 289-328; Table 3-6).

3.Variability in the relationship between OH and blood pressure In Figure 2F, several patients show high blood pressure despite low OH, whereas others exhibit low blood pressure despite high OH. This variability likely reflects the multifactorial nature of hypertension in dialysis patients, involving not only fluid overload but also vascular calcification, autonomic dysfunction, inadequate dialysis, and secondary hyperparathyroidism. I suggest adding discussion acknowledging this multifactorial background and emphasizing the clinical importance of evaluating fluid overload while also recognizing other contributors to hypertension.

Response:

Thank you for your valuable insights regarding the variability observed in Figure 2F. We fully agree that while assessing fluid overload remains clinically central to blood pressure management, hypertension in MHD patients involves multiple contributing factors. In this study, we have statistically examined the role of other relevant factors (e.g., iPTH, dialysis adequacy) and have expanded the Discussion to explicitly acknowledge this clinical complexity (Page 24, Line 407-425).

4.Stronger correlations of ECW/TBW and ECW/ICW compared with OH. It is interesting that ECW/TBW and ECW/ICW show stronger correlations with blood pressure compared to OH. These indices reflect the relative increase in ECW in the context of reduced ICW. In elderly, malnourished, or sarcopenic patients, the physiological buffer capacity against fluid overload is diminished, and even a similar OH value may exert a greater hemodynamic impact. Because OH represents an absolute excess fluid volume estimated by BCM, it may underestimate fluid overload in such populations. From this perspective, ECW/TBW and ECW/ICW may act as more sensitive indices in detecting clinically relevant fluid disturbances. I encourage the authors to expand the discussion accordingly.

Response:

Thank you for this insightful observation. As suggested, we have expanded the Discussion section to address this directly, explaining why ECW/TBW and ECW/ICW may be more sensitive markers of clinically relevant fluid disturbances than OH in MHD patients with reduced ICW (Page 23, Line 397-406).

Response to Reviewer #2

1.Timing of blood sampling, BCM, and ABPM measurements.

The timing of blood sampling, BCM measurement, and ABPM initiation is not clearly specified. Please clarify whether blood tests and BCM assessments were uniformly performed at the beginning of the dialysis week and specifically before the first dialysis session of the week. Since volume status and biochemical parameters fluctuate across the dialysis cycle, precise timing is essential for proper interpretation of the relationships between OH, laboratory data, and ABPM results.

Response:

Thank you for highlighting the need for clarity. We have revised the Methods section to specify the exact timing of all measurements. All pre-dialysis laboratory tests were performed before the first dialysis session of the week (i.e., at the end of the long interdialytic interval); the BCM assessment was also conducted at this same time point. Post-HD Scr and urea nitrogen Post-HD BUN were measured immediately after that same dialysis session. The 44-hour interdialytic ABPM was then initiated immediately after this same session (post-dialysis) and continued until the start of the next scheduled dialysis treatment (Page 7, Line 126-133; Page 7, Line 139-140; Page 8, Line 148-151).

2. Statistical methodology.

First, the method used to determine the ROC-derived cut-off values should be clearly specified. Were the cut-off values determined using the Youden Index?

Second, multivariable analyses are presented in Tables 5 and 6. However, the covariates listed in these tables do not fully match all variables described in the main text as being included in the multivariable models. Should the reader understand that all variables listed in the Analysis of factors influencing ABPM levels and circadian rhythm in the Results section were entered into the multivariable models, even if they are not displayed in Tables 5 and 6? This point requires clarification for transparency and reproducibility.

Furthermore, several closely related variables appear to have been entered simultaneously into the models, such as Pre-HD Scr and Post-HD Scr, as well as ECW, ECW/ICW, and ECW/TBW. These variables are mathematically and physiologically interdependent. How was multicollinearity assessed, and how was it addressed in the regression models? Without appropriate handling of multicollinearity, the stability and validity of the reported independent predictors may be compromised.

Response:

Thank you for these critical methodological points, which have greatly helped us improve the manuscript. In the revised manuscript, we have clarified all three issues.

First, we now explicitly state in the Statistical Analysis section that “the optimal cut-off value was determined by maximizing Youden’s index.” (Page 10, Line 196-197)

Second, regarding the variables presented in the multivariable models (Tables 5 and 6), we acknowledge that our initial manuscript did not clearly present the results of the univariate analysis. To improve transparency, we have now reformatted Tables 5 and 6 to include results from the preceding univariate screening step, and we have updated the description of the analytical sequence in the Statistical Analysis section. Furthermore, following a suggestion from another reviewer, the volume indicators TBW, ECW, and ICW have been replaced with their respective BSA-adjusted values (TBW/BSA, ECW/BSA, ICW/BSA) in all analyses (Page 10, Line 192-194; Page 17, Line 294-305; Page 19, Line 320-328).

Third, to address multicollinearity, we assessed variance inflation factors (VIF) and excluded variables with VIF ≥ 5. This ensured that correlated variables (e.g., different fluid status ratios) were not entered simultaneously; only the most representative variable from each set was retained. This clarification has been added to both the Methods and Results sections to ensure the stability and validity of the reported predictors (Page 10, Line 192-194; Page 17, Line 294-305; Page 19, Line 320-328).

Thank you again for your valuable suggestions, which have strengthened the manuscript.

3. Take-home message and clinical interpretation.

The authors’ main conclusion is that OH is associated with poor blood pressure control and abnormal ABPM circadian rhythm. As this is an observational study, causality cannot be established. However, the Discussion and Conclusion emphasize the concept of “personalized treatment strategies.” Could the authors clarify whether this concept essentially means that reducing pre-dialysis OH—that is, minimizing interdialytic weight gain—is the key practical implication of their findings? Explicitly stating this clinical message would greatly strengthen the practical relevance and interpretability of the study.

Response:

Thank you for this important insight regarding the clinical message of our study. We fully agree that clarifying the core practical implication is crucial for enhancing the translational value of our findings. In the revised Discussion and Conclusion section, we have removed the broad reference to "personalized treatment strategies" and explicitly state that a key actionable implication of our findings is that reducing OH represents the central, practical strategy for improving blood pressure control and circadian rhythm. This specific formulation translates the observational associations into a clear target for clinical action, thereby strengthening the practical guidance of our study. Thank you again for helping us improve the clinical interpretability of our work (Page 25-26, Line 442-458).

Minor Comments.

1.Please spell out the abbreviation NSDP in the Abstract.

Response:

We thank the reviewer for this observation. The abbreviation “NSDP” has now been expanded to its full form in the Abstract(Page 3, Line 43).

2. The manuscript uses multiple terms such as “abnormal circadian rhythm,” “ABPM rhythm,” and “abnormal blood pressure rhythm.” Are these terms intended to represent exactly the same concept? If so, please unify the terminology for consistency. If not, please clarify their respective definitions and distinctions.

Response:

We thank the reviewer for this note. The terminology has been unified to “circadian blood pressure (BP) rhythm ”throughout the manuscript. We have also conducted a thorough check and standardized the use of other abbreviations in the text to ensure overall consistency.

---

## [Decision Letter · Decision Letter 1]

13 Jan 2026

Bioimpedance-assessed volume overload predicts interdialytic hypertension and disrupted circadian blood pressure rhythm in maintenance hemodialysis patients

PONE-D-25-57307R1

Dear Dr. Bai,

We’re pleased to inform you that your manuscript has been judged scientifically suitable for publication and will be formally accepted for publication once it meets all outstanding technical requirements.

Kind regards,

Yosuke Yamada

Academic Editor

PLOS One

Additional Editor Comments (optional):

Reviewers' comments:

Reviewer's Responses to Questions

**Comments to the Author**

1. If the authors have adequately addressed your comments raised in a previous round of review and you feel that this manuscript is now acceptable for publication, you may indicate that here to bypass the “Comments to the Author” section, enter your conflict of interest statement in the “Confidential to Editor” section, and submit your "Accept" recommendation.

Reviewer #1: All comments have been addressed

Reviewer #2: (No Response)

2. Is the manuscript technically sound, and do the data support the conclusions?

Reviewer #1: Yes

Reviewer #2: (No Response)

3. Has the statistical analysis been performed appropriately and rigorously? 

Reviewer #1: Yes

Reviewer #2: (No Response)

4. Have the authors made all data underlying the findings in their manuscript fully available?

Reviewer #1: Yes

Reviewer #2: (No Response)

5. Is the manuscript presented in an intelligible fashion and written in standard English?

Reviewer #1: Yes

Reviewer #2: (No Response)

6. Review Comments to the Author

Reviewer #1: (No Response)

Reviewer #2: In this revised version, the authors have appropriately addressed the reviewers’ comments, and I have no further comments.

7. PLOS authors have the option to publish the peer review history of their article (what does this mean?). If published, this will include your full peer review and any attached files.

Reviewer #1: No

Reviewer #2: **Yes: **Takuya Fujimaru

---

## [Editor Report · Acceptance letter]

PONE-D-25-57307R1

PLOS One

Dear Dr. Bai,

I'm pleased to inform you that your manuscript has been deemed suitable for publication in PLOS One. Congratulations! Your manuscript is now being handed over to our production team.

Kind regards,

on behalf of

Dr. Yosuke Yamada

Academic Editor

PLOS One